# Response Characteristics of Contactless Impedance Detection (CID) Sensor on Slug Flow in Small Channels: The Investigation on Slug Separation Distance

**DOI:** 10.3390/s22228987

**Published:** 2022-11-20

**Authors:** Chenxu Wang, Junchao Huang, Haifeng Ji, Zhiyao Huang

**Affiliations:** State Key Laboratory of Industrial Control Technology, College of Control Science and Engineering, Zhejiang University, Hangzhou 310027, China

**Keywords:** contactless impedance detection (CID), sensor, slug flow, small channels

## Abstract

In recent years, CID sensors have displayed great development potential in parameter measurement of gas–liquid two-phase flow in small channels. However, the fundamental/mechanism research on the response characteristics of CID sensors is relatively insufficient. This work focuses on the investigation of the influence of separation distance between slugs on the impedance (real part, imaginary part and amplitude) response characteristics of slug flow in small channels. Experiments were carried out with the CID sensors in four small channels with inner pipe diameters of 1.96 mm, 2.48 mm, 3.02 mm and 3.54 mm, respectively. The experimental results show that for a CID sensor, the slug separation distance has significant influence on the impedance response characteristics. There is a critical value of slug separation distance. When the slug separation distance is larger than the critical value, the impedance response characteristics of each slug can be considered independent of each other, i.e., there is no interaction between the slugs. When the slug separation distance is less than the critical value, the impedance response characteristics show obvious interaction between the slugs. It is indicated that the ratios of the critical values to the pipe inner diameters are approximate 100.

## 1. Introduction

Due to the superior quality and heat transfer efficiency of small channels, the gas–liquid two-phase flow in small channels has attracted more and more attention [1,2,3,4,5]. Small channels are widely used in chemical engineering, nuclear engineering, aerospace and other industrial fields. For different application fields, the diameter definitions of the small channels are different. In the field of microscale heat transfer, the inner diameter of the small channel is defined as 1 mm to 6 mm [4]. In the field of microelectronics, the inner diameter of a small channel is defined as 200 μm to 3 mm [5]. The research on the measurement of gas–liquid two-phase flow in small channels has both industrial application significance and scientific research value [6,7,8,9,10,11]. For gas–liquid two-phase flow in small channels, slug flow is one of the most common flow patterns [6,7,8,9,10,11,12,13]. The parameter measurement research of slug flow is mostly concentrated in the conventional channels, while the parameter measurement research of slug flow in small channels is still insufficient, which needs further research [14,15].

A contactless impedance detection (CID) technique is developed from contactless the conductivity detection (CCD) technique and can realize contactless impedance measurement [16], i.e., can obtain the real part and imaginary part of impedance information simultaneously, and it is available for applying in small channels [17,18,19]. Figure 1 shows the measurement principle of CID.

As shown in Figure 1a, the CID sensor is usually composed of an AC excitation source, sensing unit and signal processing unit. The sensing unit comprises an insulated pipe, measured fluid, excitation electrode and detection electrode. As Figure 1b shows, when a CID sensor operates, the AC source generates an AC voltage excitation signal $u_{in}$ on the excitation electrode, and the AC current signal $i_{o}$ could be obtained from the detection electrode. The electrodes, insulation pipe and measured fluid are equivalent to two coupling capacitances ($C_{1}$ and $C_{2}$), and the measured fluid is equivalent to an impedance $Z_{x}$. The previous research works have verified the potential of the CID sensor on the measurement of gas–liquid two-phase flow in small channels [20,21,22,23,24,25,26], and existing CID sensors have been used in parameter measurement of gas–liquid two-phase flow in small channels, such as void fraction, velocity and so on [27,28]. However, as a relatively new technology, the fundamental/mechanism investigations on the CID sensor are limited and insufficient. Further research should be carried out to improve the measurement performance of CID sensors. For electrical sensors, such as a CID sensor, the response characteristics of a sensor is a key factor, which is closely related to the measurement performance [28,29,30,31]. By clearly revealing the response characteristics, the measurement performance of the sensor could be improved [28,29,30,31]. For the CID sensor, when a slug in the small channel flows through the different positions of the CID sensor, the electrical impedance fluctuation signal obtained by the CID sensor is a very important response characteristic, which can effectively reflect the real-time performance and the impedance sensitivity characteristic of the CID sensor. Thus, investigating the response characteristics of a single slug will be helpful to gain insight into the measurement performance of the CID sensor. Meanwhile, it is necessary to indicate that in practical application the slug flow often appears in the form of multiple continuous slugs rather than a single slug. In this case, the separation distance between slugs and the mutual influence between the slugs will affect the response characteristics of the CID sensor. Recent relevant research work is insufficient, so it is necessary to carry out research on the response characteristics of slug flow with different slug separation distances of the CID sensors.

This work aims to investigate the response characteristics of the CID sensor on slug flow in small channels, the response characteristics of a single slug and the response characteristics of slugs with different slug separation distances. The response characteristics of individual slugs are summarized, and the influence of the different slug separation distances on the response characteristics will be compared. Whether the impedance signals of two adjacent slugs are coupled (i.e., the impedance signals of two adjacent slugs affect each other and cannot be regarded as two independent signal units) when the slug separation distance is too short will be investigated. The critical value of the slug separation distance that causes the interference of the impedance signal will be recorded. The CID sensor prototypes with four pipe inner diameters (1.96 mm, 2.48 mm, 3.02 mm and 3.54 mm, respectively) will be built to carry out experiments.

## 2. Experimental Setup

### 2.1. Contactless Impedance Detection (CID) Sensor

Figure 2 shows the circuit structure diagram of the new CID sensor used in this work.

As shown in Figure 2, the impedance measurement unit can be mainly divided into three parts, the simulated inductor module, the I/V circuit and the analog phase sensitive demodulation (APSD) module.

$Z_{x}$ denotes the equivalent impedance of the measured gas–liquid two-phase flow, the impedance $Z_{x}$ can be expressed as:(1)$$Z_{x}=R_{e}+jI_{m}$$
where j is the imaginary unit, and $R_{e}$ and $I_{m}$ are the real part and imaginary part of the impedance, respectively. The inductive reactance generated by the simulated inductor is used to eliminate the capacitive reactance of the coupling capacitance, hence to realize the series resonance, which is the key to overcome the adverse effect of the coupling capacitance [32]. In order to obtain real and imaginary parts of electrical impedance, the APSD technique is introduced [33,34]. With the output signals of the APSD module $U_{0}$ and $U_{90}$, the values of real part $R_{e}$, the imaginary part $I_{m}$ and amplitude $A_{m}$ of the impedance can be obtained as:(2)$$R_{e}=-\frac{A^{2}R_{f}}{2}\frac{U_{0}}{U_{0}^{2}+U_{90}^{2}}$$
(3)$$I_{e}=\frac{A^{2}R_{f}}{2}\frac{U_{90}}{U_{0}^{2}+U_{90}^{2}}$$
(4)$$A_{m}=\frac{A^{2}R_{f}}{2}\sqrt{\frac{1}{U_{0}^{2}+U_{90}^{2}}}$$
where *A* is the amplitude of $u_{in}$, and $R_{f}$ is the feedback resistance of the I/V circuit.

### 2.2. Experimental Equipment

The response characteristics of the CID sensor can be analyzed through a large number of experiments. Figure 3 shows the diagram of experimental equipment.

The experimental device includes a fluid driving module, a high-speed camera module and a flow measurement module.

The function of the fluid drive module is to form gas–liquid two-phase flows in different states in the experimental pipe. The fluid drive part is composed of a high-pressure nitrogen tank, a water tank, a surge tank, a gas flowmeter, a liquid flowmeter and a mixer. Pressure gauges and thermometers are installed in the water tank and gas pressure stabilizing tank, respectively, to measure the air pressure and temperature in the tank. Liquid flowrate is measured and controlled by a liquid mass flowmeter (ACU10L-LC, Beijing ACCUFLOW Technology Co.,Ltd, Beijing, China). Gas flowrate is measured and controlled by a gas mass flowmeter (F-201 CB, Bronkhorst High-Tech B.V Inc., Ruurlo, Netherlands). In practical experiments, gas and water were injected continuously by a gas mass flowmeter and a liquid mass flowmeter, respectively, and by adjusting the flowrates of the gas and water, the slugs with required separate length and slug length could be obtained. To guarantee that the flow was developed to be stable, the developing length before the flow reached the measurement position (i.e., the distance between the export of the mixer and the left edge of the CID sensor) was set as 1.00 m.

The function of the high-speed camera module is to capture the dynamic image of the slug flow and obtain the slug separation distance. The module is composed of a high-speed camera, a cold light source and a computer. The high-speed camera used is the MotionXtra N-4 series produced by IDT Redlake, with a maximum image resolution of 1024 × 1024. The shooting speed can reach 3000 frames per second (FPS) at the maximum resolution.

The function of the two-phase flow parameter measurement module is to obtain the electrical impedance signal of the gas–liquid two-phase flow. The module is composed of an impedance measurement unit and a data acquisition unit. The impedance measurement unit is composed of a CID sensor and its corresponding control circuit and is used to obtain an electrical signal reflecting the impedance information of slug flow. The data acquisition unit adopts the NI9172 CompactDAQ acquisition module produced by national instruments, which is used to collect the electrical signals output by the CID sensor and transmit them to the computer for subsequent data processing. Figure 4 shows the photograph of the experimental device.

The small channels used in the experiment were horizontal glass channels with different pipe inner diameters, and the experimental temperature was 20.2 °C. Small channels with inner diameters of 1.96 mm, 2.48 mm, 3.02 mm and 3.54 mm were selected to be assembled into four sets of CID sensor prototypes, and more details of four prototypes are shown in Table 1.

## 3. Experimental Results

### 3.1. Definition Statements

The static experiments were carried out first to test the static characteristics of the CID sensors. In the static experiments, the original impedance signals (real part signal *r*_*e*_, imaginary part signal *i*_*m*_ and amplitude signal $a_{m}$ ($a_{m}=\sqrt{r_{e}^{2}+i_{m}^{2}}$)) of the channel full of water were obtained. Figure 5 shows the original impedance signals of the channel full of water obtained by the CID sensor in 10.0 s, respectively.

As shown in Figure 5, the impedance signals (*r*_*e*_, *i*_*m*_ and *a*_*m*_) are stable and have low noise level. If there is no slug, the fluid can be regarded as a pure conductance. A single slug is the basic component of slug flow, so it is necessary to investigate whether the response characteristics of CID sensor can reflect flowing characteristics of the slug flow with a single slug.

Figure 6 shows the definition of slug separation distance and $x_{slug}$ for the experiments in this part. As shown in Figure 6, the origin of the coordinate axes is selected on the left edge of the excitation electrode, then slug axial position $x_{slug}$ is defined as axial relative position between the rightmost axial position of the slug and the origin. During the experiment, the slug flowed from the excitation electrode to the detection electrode in the axial direction. When $x_{slug}$ is located on the left side of the origin, its value is specified as a negative value. When $x_{slug}$ is located to the right of the origin, its value is specified as a positive value.

Figure 7a,b show a photograph captured by the high-speed camera and impedance signals of slug flow with the slug separation distance of 9.0 cm.

As Figure 7a shows, the $x_{slug}$ is −1.8 cm in this case, and the impedance (real part, imaginary part and amplitude) signals correspond to the impedance signals at the time indicated by the arrow in Figure 7b.

In this work, the difference between the obtained impedance signals of the measured slug flow and the impedance signals of the channel full of water are used to justify whether the impedance signal is the impedance signal corresponding to a slug:
(1)Calculate the mean value of the impedance signals of the channel full of water ${\bar{s}}_{full}$
and standard deviation $\sigma_{full}$.(2)If one of the following conditions is satisfied, the impedance signal $s_{slug}\left(x_{slug}\right)$
is the impedance signal corresponding to a slug.
(5)$$\begin{array}{ll} \mathrm{Condition}\;1: & \left|s_{slug}\left(x_{slug}\right)-{\bar{s}}_{full}\right|>h\sigma_{full}\\ \mathrm{Condition}\;2: & \left|k\left(x_{slug}\right)\right|>\varepsilon \end{array}$$
where h is signal fluctuation judgement coefficient, ε is slope judgement coefficient, and the slope of the signal is defined as $k\left(x_{slug}\right)=s\left(x_{slug}-1\right)-s\left(x_{slug}\right)$.

Using the experimental data (which pipe inner diameter is 3.54 cm and slug separation distance is 63 cm) as an example, through the above separating method, the impedance signals of the slug are marked as red, and the impedance signals of the channel full of water are marked as black. Figure 8 shows the marked results.

Figure 9a–c show the impedance signals of a single slug. Slug flow of signal separation distance is 63 cm, and slug flow of signal separation distance is 9 cm. As Figure 9c shows, it is noteworthy that when signals corresponding to slugs are close, the signals have interaction. In this work, to describe the interaction of slugs, the signal correlation coefficient ρ is introduced. Thee definition of ρ is:(6)$$\rho =\frac{1}{N-1}\overset{N}{\underset{x=1}{\sum }}\frac{s_{si}\left(x\right)-\mu_{si}}{\sigma_{si}}\frac{s_{slug}\left(x\right)-\mu_{slug}}{\sigma_{slug}}$$
where $s_{si}\left(x\right)$ is the impedance signal corresponding to a single slug (i.e., shown in the curve of the red part in Figure 9a), N is the length of $s_{si}\left(x\right)$, $\mu_{si}$ and $\sigma_{si}$ are the average value and the std of the impedance signals of $s_{si}\left(x\right)$, $s_{slug}\left(x\right)$ is the single slug impedance signal corresponding to a slug flow, the interval of $s_{slug}\left(x\right)$ is intercept according to the left and right end points of $s_{si}\left(x\right)$, and $\mu_{slug}$ and $\sigma_{slug}$ are the average value and the std of the impedance signals of $s_{slug}\left(x\right)$. The total impedance information signals include the real part signal, the imaginary part signal and the amplitude signal. Thus, three signal correlation coefficients, $\rho_{re}$, $\rho_{im}$ and $\rho_{am},$ will be obtained.

As Figure 9b shows, when the slug separation distance is 63 cm, $\rho_{re}$ = 0.93, $\rho_{im}$ = 0.94 and $\rho_{am}$ = 0.93. As Figure 9c shows, when the slug separation distance is 14 cm,$\rho_{re}$ = 0.43, $\rho_{im}$ = 0.46 and $\rho_{am}$ = 0.43. In this case, the signals have interaction, and the signal correlation coefficients will decrease. Therefore, signal correlation coefficients can be used to describe whether signals have interaction.

### 3.2. Experimental Results and Discussions

The experiment was carried out by using four prototypes with different pipe inner diameters (1.96 mm, 2.48 mm, 3.02 mm and 3.54 mm, respectively). Figure 10a shows the impedance signals (*r*_*e*_, *i*_*m*_ and *a*_*m*_) of a single slug obtained by the CID sensor with pipe inner diameter of 1.96 mm when a single slug flows through the CID sensor. Figure 10b–d show the impedance signals of slug flow with different slug separation distances obtained by the CID sensor with pipe inner diameter of 1.96 mm. Figure 11, Figure 12 and Figure 13 show the impedance signals obtained by the CID sensor with pipe inner diameters of 2.48 mm, 3.02 mm and 3.54 mm, respectively.

As shown in Figure 10a, the presence of slugs leads to changes in impedance. With the slug passing the CID sensor, the changing trends of the real part, the imaginary part and the amplitude are different. For the real part, the impedance value has the trend of increasing firstly and then decreasing to the value of full of water. For the imaginary part, the impedance value has the trend of decreasing firstly and then increasing to the value of full of water. For the amplitude, the impedance value has the trend of increasing firstly and then decreasing to the value of full of water. Meanwhile, the response characteristics of the real part, the imaginary part and the amplitude also have some similarities. When the separation distance between the slug and the CID sensor is large enough, the fluctuations of the impedance signals are slight, and the impedance signals are almost the same as the signals of the channel full of water. As shown in Figure 10b, $\rho_{re}$ = 0.88, $\rho_{im}$ = 0.99 and $\rho_{am}$ = 0.88. The signals of the slug flow are independent of each other, i.e., there is no interference phenomenon of corresponding impedance signals. It can also be seen that for the real part, the imaginary part and the amplitude signal, the separation distances are smaller than the actual slug separation distance. As shown in Figure 10c, with the decrease of the slug separation distance, the signal separation distances decrease, and
$\rho_{re}=0.94$, $\rho_{im}=0.99$ and $\rho_{am}=0.94$. There is still no interference phenomenon of corresponding impedance signals. The impedance response characteristics of each slug can be regarded as independent response characteristics and have no effect on each other. It is necessary to point out that there is a difference between the signal correlation coefficient in Figure 10b and that in Figure 10c, which may be due to the slight difference in the slug length between the two experiments. As shown in Figure 10d, with the further decrease of the slug separation distance, the signal correlation coefficients decrease sharply, where $\rho_{re}=0.30$, $\rho_{im}=0.49$ and $\rho_{am}=0.30$. Interference phenomenon of corresponding impedance signals occurs, and the impedance response characteristics of each slug cannot be regarded as independent response characteristics.


By comparing Figure 10, Figure 11, Figure 12 and Figure 13, in a small channel, with the decrease of slug separation distance, the corresponding slug impedance signals will appear interactive and will not be regarded as independent slug signals, and signal correlation coefficients will decrease due to the interaction phenomenon of signals, which exist similar to the experiments of CID sensors with different pipe inner diameters.

In this work, critical separation distance values of real part $d_{re}$, the separation distance critical values of imaginary part $d_{im}$ and separation distance critical values of amplitude $d_{am}$ are introduced in order to better describe response characteristics. The definition of the critical separation distance value is the maximum value of slug separation distance to avoid the interference phenomenon of corresponding impedance signals. Table 2 shows the experimental results.

As shown in Table 2, for the four prototypes involved in the experiment, the critical values ($d_{re}$, $d_{im}$ and $d_{am}$) of slug separation distance increase with the increase of pipe inner diameter. Therefore, the impedance signals (*r*_*e*_, *i*_*m*_ and *a*_*m*_) obtained by the CID sensor with the larger pipe inner diameter are more likely to be coupled. From the experiments of the four pipe inner diameters, it is found that the critical values ($d_{re}$, $d_{im}$ and $d_{am}$) are different under the same pipe inner diameter, where the critical values of the real part and the amplitude are slightly different, and the critical value of the imaginary part is slightly smaller than that of the real part and the amplitude. As experimental results show, the ratios of the critical values ($d_{re}$, $d_{im}$ and $d_{am}$) to the pipe inner diameters are approximately 100.

## 4. Conclusions

This work focuses on the response characteristics of a CID sensor for slug flow in small channels. With experiments, the response characteristics of the CID sensor for the single slug and slug flow with different slug separation distances were investigated and concluded. The response characteristics of the CID sensor for slug flow with different slug separation distances were investigated by the CID sensor with the pipe inner diameters of 1.96 mm, 2.48 mm, 3.02 mm and 3.54 mm, respectively. The experimental result shows that the changing trends of the real part, the imaginary part and the amplitude with the slug flow are different. Meanwhile, there also exist similarities in the response characteristics of different parts of the impedance. The response characteristics of the CID sensor can effectively reflect the flowing characteristics of the slug flow.

The experimental results also show that the slug separation distance has significant effects on the response characteristics of the CID sensor. For each small channel, there exists a critical separation distance value, and the ratios of the critical values to the inner pipe diameters are approximately 100. However, the ratio of slug separation distance to pipe inner diameter is generally less than 100 for the actual slug flow, so it is necessary to consider the effect of slug separation distance on the response characteristics of the CID sensors. When the slug separation distance is larger than the critical separation distance value, the corresponding impedance signals of slugs can be regarded as independent slug signals, i.e., there is no interaction between the slugs. In this case, the relatively large signal correlation coefficient indicates that the slug signal obtained from the slug flow has a high similarity to the single slug signal. When the slug separation distance is less than the critical separation distance value, the corresponding slug impedance signals show obvious interaction and cannot be regarded as independent slug signals. In this case, the signal correlation coefficients decrease sharply. For a CID sensor, critical separation distance values of the real part and the amplitude are almost the same and are larger than that of the imaginary part. For the CID sensors with different pipe inner diameters, the increasing of pipe inner diameter will lead to the increasing of the critical separation distance values of the real part, the imaginary part and the amplitude.

This work focuses on the investigation of the effects of slug separation distance and pipe inner diameter on the response characteristics of the new CID sensor for slug flow. The research results can provide useful reference for further parameter measurement and sensor development. Except for these aspects, many other factors, such as the effects of slug length, electrode length, electrode spacing and so on, are also worth investigating on the response characteristics of the new CID sensor. In addition, this work provides experimental results as references, and the relevant mechanism analysis will be further explored.

## Figures and Tables

**Figure 1 sensors-22-08987-f001:**
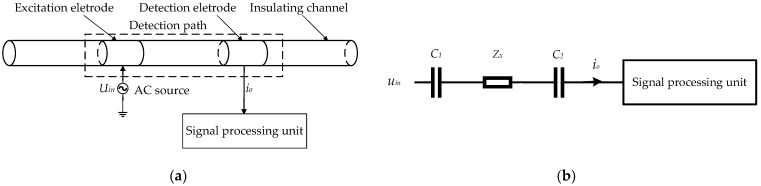
Measurement principle of CID: (**a**) construction of a typical CID sensor; (**b**) simplified circuit of the CID sensor.

**Figure 2 sensors-22-08987-f002:**
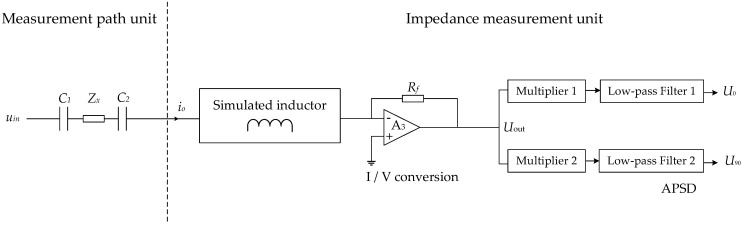
Principle circuit of the new CID sensor.

**Figure 3 sensors-22-08987-f003:**
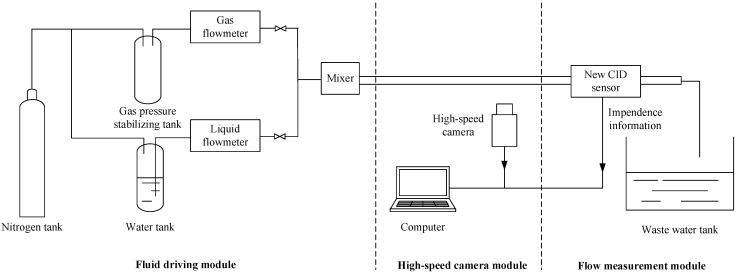
Diagram of the experimental setup.

**Figure 4 sensors-22-08987-f004:**
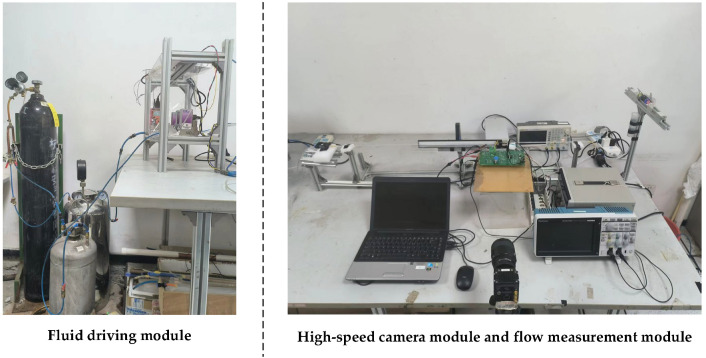
Photograph of the experimental device.

**Figure 5 sensors-22-08987-f005:**
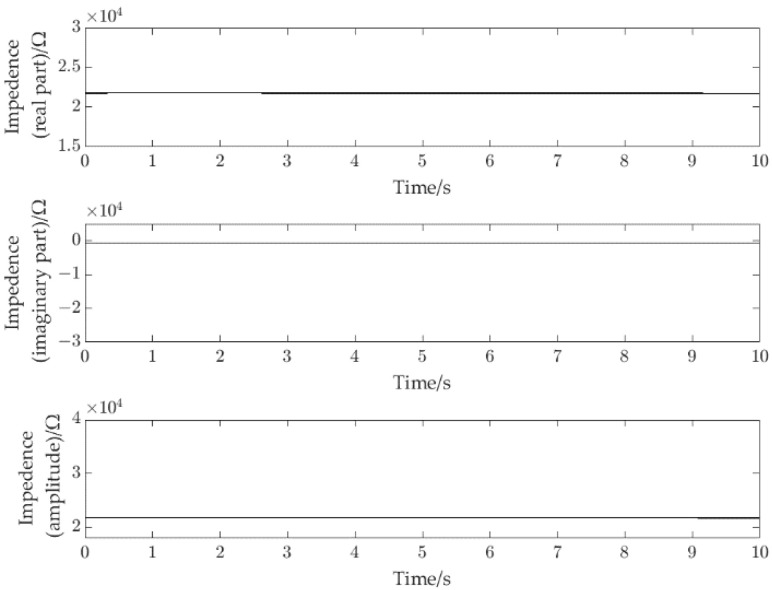
Impedance signals of full of water, i.d. = 2.48 mm.

**Figure 6 sensors-22-08987-f006:**
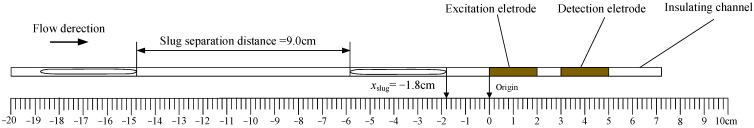
Diagram of relative position of the slug and CID sensor in the axial direction.

**Figure 7 sensors-22-08987-f007:**
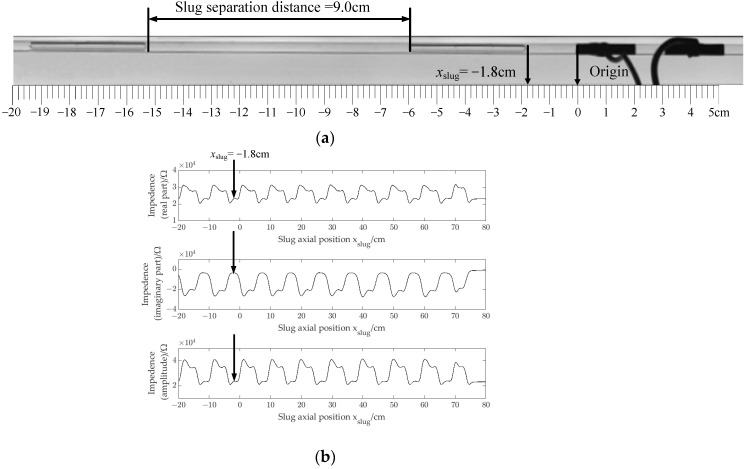
Photographs and impedance signals: (**a**) photographs; (**b**) impedance signals.

**Figure 8 sensors-22-08987-f008:**
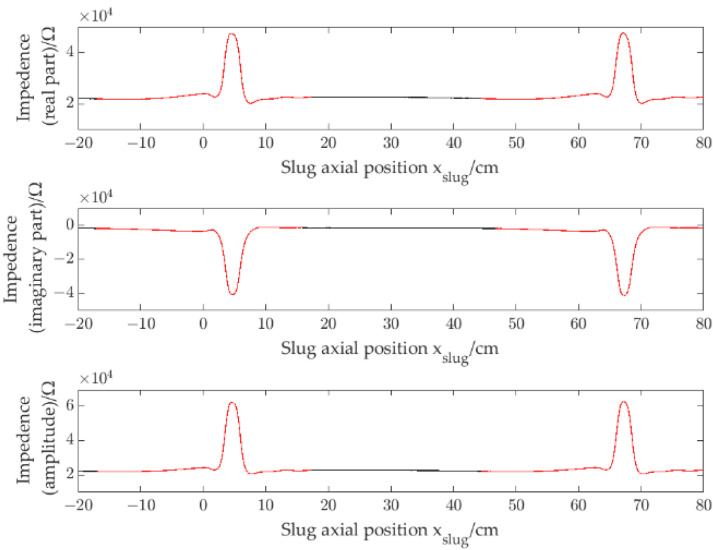
The marked results of impedance signals.

**Figure 9 sensors-22-08987-f009:**
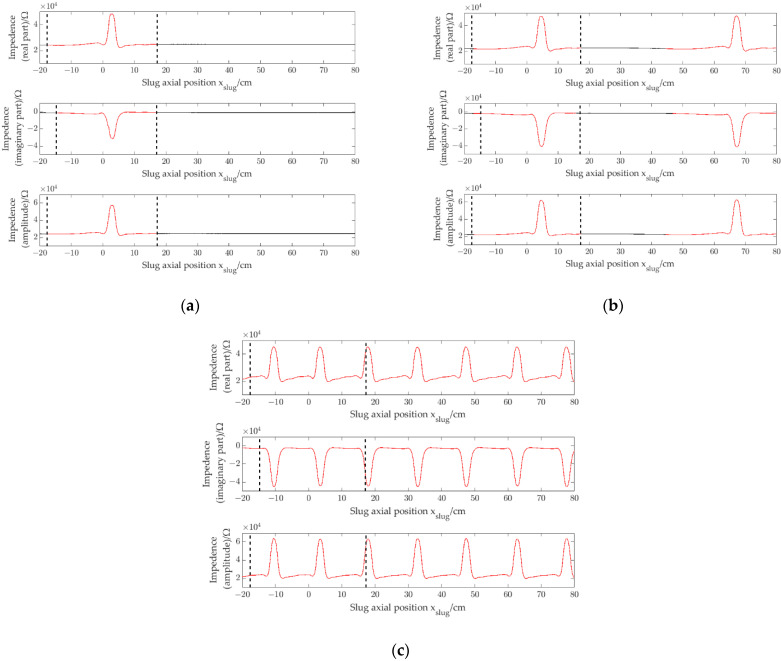
Impedance signals of slug flow, i.d. = 3.54 mm: (**a**) single slug; (**b**) signal separation distance is 63 cm; (**c**) signal separation distance is 14 cm.

**Figure 10 sensors-22-08987-f010:**
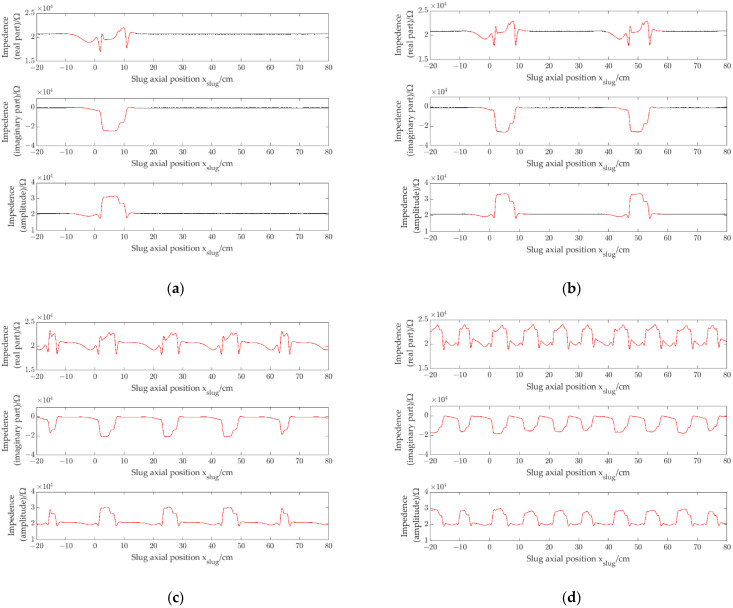
Impedance signals of slug flow, i.d. = 1.96 mm: (**a**) single slug; (**b**) signal separation distance is 44 cm ($\rho_{re}=0.88$, $\rho_{im}=0.99$ and $\rho_{am}=0.88$); (**c**) signal separation distance is 22 cm ($\rho_{re}=0.94$, $\rho_{im}=0.99$ and $\rho_{am}=0.94$); (**d**) signal separation distance is 11 cm ($\rho_{re}=0.30$, $\rho_{im}=0.49$ and $\rho_{am}=0.30$).

**Figure 11 sensors-22-08987-f011:**
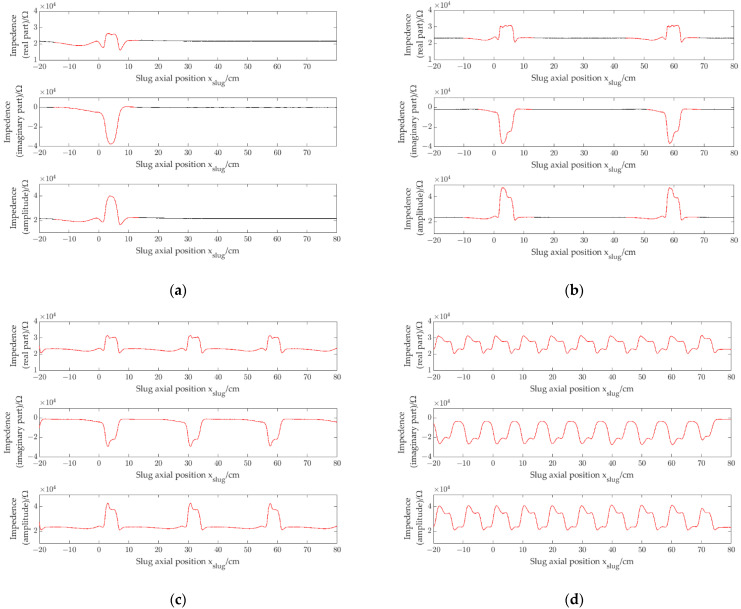
Impedance signals of slug flow, i.d. = 2.48 mm: (**a**) single slug; (**b**) signal separation distance is 56 cm ($\rho_{re}=0.91$, $\rho_{im}=0.95$ and $\rho_{am}=0.91$); (**c**) signal separation distance is 27 cm ($\rho_{re}=0.89$, $\rho_{im}=0.95$ and $\rho_{am}=0.89$); (**d**) signal separation distance is 9 cm ($\rho_{re}=0.38$, $\rho_{im}=0.47$ and $\rho_{am}=0.38)$.

**Figure 12 sensors-22-08987-f012:**
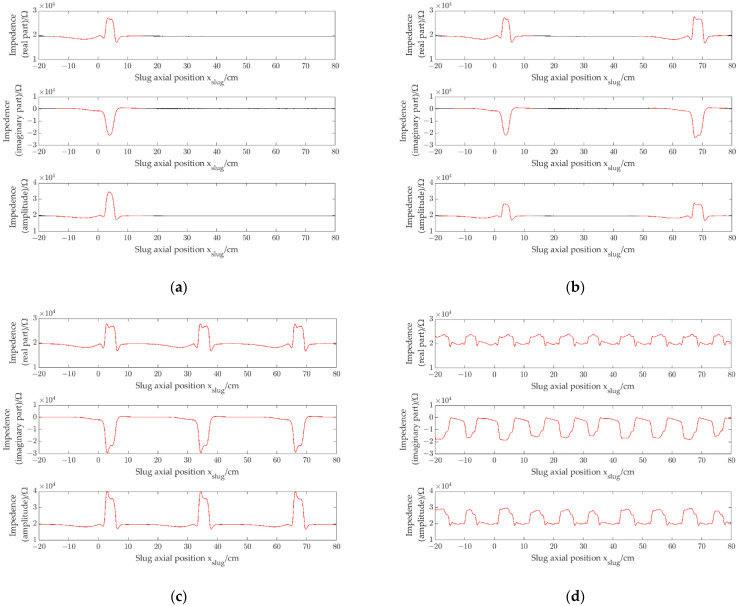
Impedance signals of slug flow, i.d. = 3.02 mm: (**a**) single slug; (**b**) signal separation distance is 64 cm ($\rho_{re}=0.95$, $\rho_{im}=0.98$ and $\rho_{am}=0.95$); (**c**) signal separation distance is 31 cm ($\rho_{re}=0.94$, $\rho_{im}=0.96$ and $\rho_{am}=0.94$); (**d**) signal separation distance is 11 cm ($\rho_{re}=0.39$, $\rho_{im}=0.53$ and $\rho_{am}=0.39)$.

**Figure 13 sensors-22-08987-f013:**
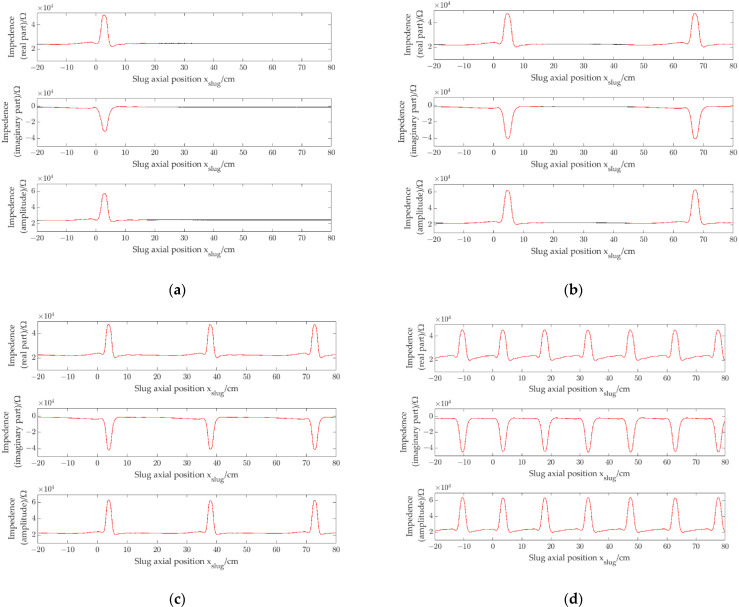
Impedance signals of slug flow, i.d. = 3.54 mm: (**a**) single slug; (**b**) signal separation distance is 63 cm ($\rho_{re}=0.93$, $\rho_{im}=0.94$ and $\rho_{am}=0.93$); (**c**) signal separation distance is 34 cm ($\rho_{re}=0.95$, $\rho_{im}=0.97$ and $\rho_{am}=0.95$); (**d**) signal separation distance is 14 cm ($\rho_{re}=0.43$, $\rho_{im}=0.46$ and $\rho_{am}=0.43$).

**Table 1 sensors-22-08987-t001:** The detailed information of the CID sensors.

	Inter Diameter (mm)	Outer Diameter (mm)	Electrode Length (mm)	Electrode Spacing (mm)
Prototype I	1.96	3.94	20.07	10.05
Prototype II	2.48	4.50	20.04	10.04
Prototype III	3.02	4.98	20.05	10.03
Prototype IV	3.54	5.48	20.04	10.00

**Table 2 sensors-22-08987-t002:** Critical separation distance values of different inner diameters.

Inter Diameter (mm)	$d_{re}$ (cm)	$d_{im}$ (cm)	$d_{am}$ (cm)	$d_{re}$/i.d.	$d_{im}$/i.d.	$d_{am}$/i.d.
1.96	22	19	23	112	97	112
2.48	27	26	27	108	104	108
3.02	31	30	31	102	99	102
3.54	35	34	31	98	96	98

## Data Availability

The data presented in this study are available on request from the corresponding author. The data are not publicly available due to policy reasons.

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
