# Peer review of "Response Characteristics of Contactless Impedance Detection (CID) Sensor on Slug Flow in Small Channels: The Investigation on Slug Separation Distance"

_sensors, 2022, doi:10.3390/s22228987_

Round 1

Reviewer 1 Report

Dear authors:

The paper is relatively well written and organized, although I think that some parts could be synthesized, especially the Introduction section where some words and sentences become repetitive. The experimental setup and the corresponding measurement results are clear and well presented. However, I have major doubts about the the novelty of the work and the relevance of the experimental results.

On this basis, I think that two aspects  should be clarified to consider the paper suitable for publication:

1) On the one hand, the use of CID sensors for slug flow measurements has been adressed in other works of the authors, like [15], [26] or the following reference omitted in this paper:

Z. Guo, J. Huang, Q. Huang, Y. Jiang, H. Ji and Z. Huang, "New Contactless Velocity Measurement Sensor for Bubble/Slug Flow in Small Scale Pipes," in IEEE Access, vol. 8, pp. 198035-198046, 2020, doi: 10.1109/ACCESS.2020.3034695.

Hence, I assume that the novelty of the work entirely relies on the study of a flow with different slug separations. It should be emphasized which parts of the work have been already addressed in previous publications and which ones are completely original.

2) The conclusions of the paper are vague and their relevance is not clear enough in my opinion. For example, the authors draw the following conclusion, which sounds quite obvious for me:

"When the slug separation distance is larger than the critical separation distance value, the corresponding impedance signals of slugs can be regarded as independent slug signals, [···] When the slug separation distance is less than the critical separation distance value, the corresponding slug impedance signals show obvious interaction and cannot be regarded as independent slug signals at all."

Thus, the paper contributions and the relevance of the obtained experimental results must be further justified.

Author Response

The response could be found in the appendix "Response to review 1".

Reviewer 2 Report

This paper demonstrates an important factor on the performance of impedance detector/meter for measuring slug two-phase flow. It can be a good reference in terms of optimized design of impedance meter. However, a series of key information are missing in the text that prevents the readers to understand the implementation of the experiment and place trust on the conclusion of the paper. Therefore, the paper can only be accepted after necessary revisions. The detailed comments are listed as follows.

It is recommended that in the introduction part, the authors state the general application field. For different application fields, the definition of small diameter pipe/tube can be quite different. The range of small diameter pipe adopted in this article is referred to only one article that is specific to compact heat exchangers. This small diameter pipe range, 1 mm to 6 mm, is not applicable to other engineering fields, such as chemical engineering, pipeline engineering, and nuclear engineering.

In page 2 and line 50, the author stated there is a potential of the CID sensor in measuring the two-phase flow, and this is a relatively new technology. Per the reviewer’s knowledge, the impedance-based measurement on two-phase flows has already been well-developed in the past a century year. Unless the author can elaborate what are the existing challenges for CID sensor to be applied in small channel two-phase flows, the statement is not true. They might be stated in the citations 16-22, but it should be explicit in the literature review section, as it is quite crucial to show the originality of this paper.

In terms of experiment operation, are you continuously injecting gas and water? Given the pipe diameter is small, and the gas and water flow rates can be quite small and requires fine control on the flow rate. This is quite crucial to control the separate length and slug length, since from the results, your experiment can realize accurate control of the slug separate distance and slug length. Please provide more info on how you are able to realize it.

Since you mix the two fluids before injecting into the pipe, what is the developing length before flow reach your measurement position?

CID sensor can be sensitive to the conductivity and contamination of the water? How to maintain a stable fluid condition given you are using tap water?

Imaging: given you rely on image to measure the lengths, have you performed calibration for the distortion due to the camera lens? In figure 7, the ruler is placed after process the image, but does not reflect the calibration of the image distortion.

In the discussion on the interference phenomenon of impedance signal due to the decreasing of slug separation distance (page 11), the authors did not provide a method or criterion on determining whether an interference exists or not. In other words, there is no rigorous basis for authors to state “obvious interference phenomenon” in figure 9d. It is not convincing to subjectively judging the interference by simply viewing the shape of the impedance signals. There can be many and more factors that alter the shape of the impedance signals, such as the slug shape, liquid conductance and contamination conditions. As provided in the paper, these factors were not strictly controlled in the experiment. The author should provide a more explicit criteria or method for determining the interference phenomenon.

Author Response

The response could be found in the appendix "Response to review 2".

Round 2

Reviewer 1 Report

Dear authors,

thank you for taking my previous comments into consideration.

On the one hand, the conclusions are clearer than in the previous manuscript version, i.e., the authors have estimated the value of the ratio between slug separation and pipe inner diameter (100) for allowing the detection of individual slugs with no interference.

On the other hand, the conclusions are purely empirical and not supported by any theoretical analysis, leading to some confusing points. In particular, the authors claim (line 127):

"In practical experiments, gas and water were injected continuously by a gas mass flowmeter and a liquid mass flowmeter respectively. And by adjusting the flowrates of the gas and water, the slugs with required separate length and slug length could be obtained."

However, the authors draw this conclusion about the experimental results (line 278):

"And it is necessary to indicated that, compared Figure 10b and 10c, although the distance of Figure 10b is larger than that of 10c, the signal correlation coefficient of Figure 10b is smaller than that of 10c, that may due to the slug length difference of ¿that? two experiments."

Hence:

-   Is this a contradictory result?

-   If this is an unexpected result, are the authors justifying this behavior by assuming the slug length is not (maybe) properly controlled in practice? Hence, is the reproducibility of the experiment questionable?

In this context, the following doubt comes to me: A ratio of 100 between slug separation and inner diameter has been obtained, but:

-  Does this ratio fullfill for any scenario or it may depend on other factors (such as slug length)? How? Have it been checked by the authors?

- Is this conclusion supported by any theoretical basis?

All the aforementioned doubts should be clarified by the authors.

Author Response

The response could be found in the appendix "Response to review 1(Round 2)".

Reviewer 2 Report

The authors have addressed the issues in the older draft. There is no further comment and the revised manuscript can be accepted.

Author Response

Thanks for your recognition and help.